# Infant Care Practices, Caregiver Awareness of Safe Sleep Advice and Barriers to Implementation: A Scoping Review

**DOI:** 10.3390/ijerph19137712

**Published:** 2022-06-23

**Authors:** Roni Cole, Jeanine Young, Lauren Kearney, John M. D. Thompson

**Affiliations:** 1School of Nursing, Midwifery and Paramedicine, University of the Sunshine Coast, Sippy Downs, QLD 4556, Australia; lauren.kearney@uq.edu.au; 2Women’s and Children’s Service, Sunshine Coast Hospital and Health Service, Birtinya, QLD 4575, Australia; 3Sunshine Coast Health Institute, Birtinya, QLD 4575, Australia; 4Queensland Child Death Review Board, Brisbane, QLD 4000, Australia; 5School of Nursing, Midwifery and Social Work, University of Queensland, Herston, QLD 4006, Australia; 6Department of Paediatrics: Child and Youth Health, Faculty of Medical and Health Sciences, University of Auckland, Grafton, Auckland 1023, New Zealand; j.thompson@auckland.ac.nz

**Keywords:** Sudden Unexpected Death in Infancy (SUDI), sudden infant death syndrome (SIDS), sleep-related infant mortality, infant care practices, safe sleeping advice, health promotion, public health campaign

## Abstract

Modifiable infant sleep and care practices are recognised as the most important factors parents and health practitioners can influence to reduce the risk of sleep-related infant mortality. Understanding caregiver awareness of, and perceptions relating to, public health messages and identifying trends in contemporary infant care practices are essential to appropriately inform and refine future infant safe sleep advice. This scoping review sought to examine the extent and nature of empirical literature concerning infant caregiver engagement with, and implementation of, safe sleep risk-reduction advice relating to Sudden Unexpected Deaths in Infancy (SUDI). Databases including PubMed, CINAHL, Scopus, Medline, EMBASE and Ovid were searched for relevant peer reviewed publications with publication dates set between January 2000–May 2021. A total of 137 articles met eligibility criteria. Review results map current infant sleeping and care practices that families adopt, primary infant caregivers’ awareness of safe infant sleep advice and the challenges that families encounter implementing safe sleep recommendations when caring for their infant. Findings demonstrate a need for ongoing monitoring of infant sleep practices and family engagement with safe sleep advice so that potential disparities and population groups at greater risk can be identified, with focused support strategies applied.

## 1. Introduction

Reported causes for post-neonatal deaths are vast, including birth defects, infections and accidents; however, Sudden Unexpected Death in Infancy (SUDI) remains the largest contributor to post-neonatal mortality in most developed countries [1]. Throughout the global SUDI research community, there has been increasing recognition that sleep-related infant mortality incidence has plateaued over the past two decades, and great efforts are being taken to understand where significant issues persist [2,3,4]. Concomitantly, there has been a rapid upsurge in research and publications exploring the intersect between safe infant sleep advice, promulgated in SUDI risk-reduction campaigns and the effective and consistent translation of this advice into safe infant care practices adopted by families. SUDI remains the largest contributor to post-neonatal mortality in most developed countries [2].

Understanding caregiver receipt of, and engagement with, public health messages and identifying trends in contemporary infant care practices are both essential components required to appropriately inform and refine future safe sleeping recommendations. Moreover, understanding factors that influence caregiver choice and possible barriers encountered when implementing advice require innovative strategies that support families. Improving the uptake of safe sleep advice by families with young infants, especially those at greater risk, has potential to prevent infant deaths.

## 2. Review Methodology

A scoping review suitably examines the extent, range and nature of a research activity. It is a useful way of mapping fields of study where the area is complex and where it may be difficult to visualise the range of material available [5]. A scoping review requires a rigorous, structured and transparent methodological approach in order to maximise the capture of relevant information, while providing reproducible results and ensuring findings are trustworthy [6]. As scoping review methodology does not follow a formal process of methodological appraisal, or study quality critique [5], identified gaps in research knowledge related to the topic of enquiry being explored may indeed be greater than relevant content identified.

To provide structure to the review, the scoping review approach described by Arksey and O’Malley [5] and the Joanna Briggs Institute methodology for scoping reviews [7] was used. This scoping review framework involves a five-step approach: (1) identifying the research question; (2) identify relevant studies; (3) study selection; (4) charting the data and (5) collating, summarising and reporting results, with an optional sixth-step, consultation [5]. This approach allows for the inclusion of varied methodologies in order to more fully and deeply understand what is known about a particular subject [7]. It is argued the scoping approach enhances rigour, thus reducing biased searches, and provides focus and boundaries for the review. The scoping review method has been used successfully for several recent healthcare literature reviews [8,9,10].

## 3. Review Aim

The purpose of this literature review was to explore the breadth and extent of literature that relate to family engagement with infant safe sleep public health messages, mapping and summarising the identified contemporary empirical primary research studies.

The following research question was formulated to address the literature review objectives and target outcomes of interest: What is known in the peer-reviewed literature about contemporary primary infant caregivers’ awareness of current safe sleep public health messages, the challenges they encounter when implementing this advice and the prevalence of practices used in the home environment related to safe sleep advice?

## 4. Search Strategy

Search terms were developed by and agreed upon by the research team. Databases searched included Scopus, Medline, EMBASE, PubMed, CINAHL, Health Source: Nursing/Academic Edition and Maternity and Infant Care (Ovid), along with Google Scholar and secondary hand-searches of reference lists. Key search terms are presented in Table 1. Searches were limited to the English language and publication dates between January 2000 and May 2021.

### Eligibility Criteria

Literature that measured or focused on specific dimensions of infant safe sleep advice (e.g., implementation prevalence, awareness of current program messages) were included in the review. Peer-reviewed journal articles were included if they were published between the period of January 2000–May 2021. This timeframe was chosen as the incidence of sudden infant mortality largely began to plateau at the turn of the century among many western countries, with little change in rates since. While grey literature is often included in scoping reviews, due to the proliferation of safe sleep message evaluations from the caregiver perspectives since 2000, peer-reviewed literature was the focus for this review.

Broadly, eligibility criteria aimed to identify empirical literature that reported original peer-reviewed research exploring caregiver engagement with current SUDI risk reduction messages, describing at least one of the review’s focused outcome measures (see Table 2).

## 5. Screening and Source Selection

Following the search, all identified citations were collated and uploaded into EndNote (Clarivate Analytics, Philadelphia, PA, USA). Duplicates were removed. Titles and abstracts were screened against the established eligibility criteria. This was followed by the retrieval of the full-text article of any potentially relevant sources. These were assessed in detail to determine if papers met eligibility by two reviewers (R.C., J.Y.). A flow diagram outlining the selection process is provided in Figure 1.

## 6. Data Extraction and Charting

Data were extracted from included papers and entered into a Microsoft Excel spreadsheet. The following data items were extracted: title; journal; authors; year of publication; geographical location of study; study aim; sample size; study period; participant characteristics, such as infant age and type of caregiver(s) (e.g., mother, grandmother); study design/methods and key findings of target outcome variables. The data extraction form is provided in Appendix A: Overview of scoping review articles. The table provides an overview of the data extracted from each of the included papers that met eligibility criteria.

## 7. Findings and Analysis

### 7.1. Eligible Study Characteristics

#### 7.1.1. Study Designs

Most studies (*n* = 98, 72%) utilised an observational study design, with cross-sectional surveys the most often used method, being administered either online, via paper questionnaire or via telephone (*n* = 78). Some observational studies also reported using a prospective (*n* = 14) or retrospective (*n* = 5) approach. Interviews or focus groups collecting qualitative data were common data collection methods (*n* = 21). Longitudinal study designs using either surveys or interviews were reported for 22 studies. The remaining papers reported using mixed methods (*n* = 12), a two-stage clustered design (*n* = 2), a qualitative descriptive social media content analysis study design, focused ethnography, a retrospective electronic medical records chart review or a sentinel data collection design.

#### 7.1.2. Study Country of Origins

Papers reported data from a total of 26 countries, with three papers reporting data collected internationally. The United States (US) had the greatest representation, with 48% (*n* = 66) of papers reporting findings from participants living in the US, followed by Australia (*n* = 12), New Zealand (*n* = 10), the United Kingdom (UK) (*n* = 9), Canada (*n* = 4) and Sweden (*n* = 4). Israel, Turkey and Brazil were represented in three papers each. Japan, Saudi Arabia and Iran had two papers each. The remaining countries (Croatia, Germany, India, Ireland, Jordan, Malaysia, Mexico, Netherlands, Norway, Portugal, Russia, Spain, Thailand and the United Arab Emirates (UAE)) were represented in only one study each. There was one paper where the country demographics were not recorded, as the sample participants were from a Facebook mother’s group.

#### 7.1.3. Study Data Collection Periods

All articles included in this review were published between 2000 and 2021. Of the 137 papers, 21 did not report the time period in which data collection occurred. For the remaining 116 papers, findings were reported on data collected between 1993 and 2019. According to the final collection date reported, only 48 papers (35%) analysed any data collected during the last decade.

#### 7.1.4. Study Participants

Eligibility criteria for inclusion of studies for this review specified participants were to be primary infant caregivers. Papers reported data from caregivers who were mostly mothers; however, for 36 papers, some fathers were also reported as informants, and 10 papers indicated that some infant caregiver participants included grandmothers. The majority of studies collected data from caregivers whose infants were less than 6-months (*n* = 83, 61%). Ten papers included data where some participant infants were older than 12-months, and a further five papers did not state the ages of the child.

### 7.2. Analysis and Discussion of Target Outcome Measures

Key data for each of the review’s target outcome measures were extracted. Summaries of notable findings from key papers are presented below. Table 3 presents an overview of the target review outcome measures (studies that measured prevalence data related to key modifiable infant sleep practices associated with SUDI risk, safe sleep campaign message awareness and knowledge and caregiver challenges and barriers with advice implementation). Infant sleep position was the most investigated modifiable safe sleep prevalence outcome measure (*n* = 105, 77%). This was closely followed by surface sharing (*n* = 85, 62%). Only six studies [11,12,13,14,15,16] reported findings on all the target prevalence outcome measures identified to be explored in this review. An additional nine studies [17,18,19,20,21,22,23,24,25] reported prevalence findings for all except one of the target prevalence outcome measures (‘bed type/sleep surface’ or ‘items in sleep space’ were the two target outcome measures not reported in these nine studies). Ten papers [26,27,28,29,30,31,32,33,34,35] did not report any measured prevalence data, with the primary aim of these papers being to explore caregiver knowledge or awareness of current safe sleep advice messages or challenges families encounter when implementing safe sleep advice.

### 7.3. Prevalence of Key Modifiable Practices

Findings have been structured using the current Australian Safe Sleep campaign messages [148], which are supported internationally [149] and focus on a synthesis of key findings, consistent with the aims of a scoping review [5]. Please refer to Appendix A for detailed study information, including study population, sample size, study design and key target outcome findings.

#### 7.3.1. Position Infant Placed to Sleep

The substantial reduction in SUDI incidence in the early 1990s has been directly associated with the widespread ‘Back to Sleep’ campaigns adopted by many countries [150,151,152]. Avoiding all non-supine sleep positions continues to be strongly advised throughout safe sleeping and risk reduction campaigns internationally [153,154,155]. Despite this advice and the known increased risk of SUDI, non-supine sleep positioning continues to be employed by caregivers with young infants.

The most common modifiable infant care practice explored was the position parents used for placing baby to sleep (*n* = 101 studies). Prevalence changed markedly within population groups sampled. Rates were reported as high as 96.7% from a Japanese cohort [85] of parents who reported that they exclusively used the supine position, to only 48.9% within a US cohort [125].

The incidence of non-supine sleep positioning among infants in studies that measured and compared prevalence as infants grew reported an increase in prone sleeping over time. Corwin et al. [63] found that at one-month-old, 12% of infants were placed prone, increasing to 20% at three-months-old. A recent Swedish study reported similar findings, where 16% of infants were placed in a non-supine position; by 3 to 5 months of age, this had risen to 26% [133]. Hutchison et al. [90] and Shields et al. [21] also both reported 0% of infants placed prone at 6 weeks and 4 weeks, respectively; however, Hutchison and colleagues [90] found that by 4 months, 3.3% of infants were placed prone, increasing to 5.6% by 8-months-old, and Shields et al. [21] reported the prevalence of prone sleeping rising to 11.6% by 6 months of age. Inbar et al. [93] also found the incidence of prone sleeping to increase from 12.4% for infants aged 0–3 months to 17.6% by 3–6 months of age.

Bombard and colleagues [47] compared the prevalence of a non-supine sleep positioning of 2- to 6-month-old infants over the period from 2009 to 2015, reporting a decrease in incidence from 27.2% to 19.4%, respectively. Similar findings were reported by Sperhake et al. [132] within a German population, where the rate of prone positioning declined from 8.1% in 1996 to 3.5% in 2006 among infants approximately 14-weeks-old. Hutchinson et al. [19] also found more mothers usually used the supine sleep position in 2013 compared with 2005. Salm Ward et al. [125], however, found that from 2004 to 2013, there was a decrease in the proportion of mothers, with babies aged 2- to 6-months, who were reporting that they most often put their infant to sleep on their back (59.7% in 2004 to 48.9% in 2013).

Several studies examined the prevalence of caregivers using a non-supine sleeping position when it was not usual or regular practice for their infant; that is, the position was used some of the time, occasionally, infrequently, or never. Smylie et al. [131] found that 23% of mothers reported putting their infant in a non-supine sleep position at some time during the first 4 months. Colson et al. [58] had similar findings, with 34% reporting ‘ever’ placing their infant prone. Two studies [49,69] assessed caregivers who reported not always, or infrequently, using the supine sleep position, reporting a prevalence of 41.7% and 42.3%, respectively. Anderson and colleagues [38] reported disparities among two geographically contrasting social classes, a socioeconomically deprived inner-city estate compared to an affluent suburban estate, when assessing infant sleeping position, finding that at 6 to 8-weeks postpartum, 29% of inner-city infants occasionally slept prone compared to 19% of suburban infants. A study investigating the difference in infant care practices between preterm and term infants found infants who are at an increased risk of SUDI due to infant factors, such as premature birth, very low birthweight and in-utero smoke exposure, are less likely to be placed supine to sleep [156].

#### 7.3.2. Infant Smoke Exposure

Tobacco smoke exposure continues to be one of the most important modifiable risk factors in reducing the risk of SUDI [150]. Globally SUDI risk reduction campaigns advise to keep a baby in a smoke-free environment [153,154,155,157]. However, findings from this review indicate that maternal smoking and smoke exposure both during pregnancy and postpartum continues to remain high among many sampled cohorts, despite the advice to avoid infant smoke exposure, before birth and after.

Sixty-two papers reported data examining the prevalence of infant smoke exposure within their sampled population cohorts. Studies reported in-utero smoke exposure and/or environmental smoke exposure. Environmental exposure included either maternal postpartum smoking or exposure from other household member(s) who smoked. Several studies found that maternal smoking rates are often lower than paternal or household smoking rates, with studies reporting maternal smoking rates between 2–4-times lower than paternal or household smoking rates (18.4% of fathers compared to 9.8% of mothers [12]; 19.2% of fathers compared to 10.2% of mothers [44]; 34% of other household members versus 17% of mothers [79]; 12.4% of fathers compared to 5.8% of mothers [16]; 7.6% of fathers compared to 3.5% of mothers smoked and 31.6% of fathers versus 10.2% of mothers used moist tobacco [113]). In a sample of Japanese infant caregivers, Sawaguchi et al. [85] found that the incidence of fathers smoking (47.9%) was much higher than mothers who smoked (9.4%). This finding was consistent among a more contemporary Japanese cohort, where paternal and maternal smoking habits at one-month postpartum were 41.7% and 2.1%, respectively [85]. These authors also found that mothers who smoked tobacco were more likely to feed their infant formula rather than breastmilk and that the absence of paternal smoking habits was a significant predictive factor for maternal smoking cessation [85].

Some studies also evaluated the proportion of infants who shared a sleeping surface with a mother who smoked, a practice reported to significant increase in the risk of infant death, as demonstrated in a meta-analysis with an odds ratio of 6.27 (95% CI: 3.94–9.99) [158]. A recent US study [87] found that households with tobacco exposure compared to smoke-free households were more likely to adopt practices which could place infants at a greater risk of sudden infant death, with supine positioning used less (75% vs. 88%), and bed-sharing (62% vs. 44%) and soft bedding more likely (50% vs. 32%). A study investigating bed-sharing practices among postpartum smoking mothers demonstrated 18.8% reported always bed-sharing; 12.6% almost always bed-share and 45.1% reported sometimes bed-sharing, with only 23.6% never bed-sharing [99]. A recent Norwegian study found that more than half (51.4%) of mothers who smoked routinely bed-shared at night [113].

#### 7.3.3. Items in an Infant’s Sleeping Space

Internationally safe sleep advice recommends avoiding soft, bulky or loose bedding or items in an infant’s sleeping space as they could inadvertently occlude an infant’s airway [153,154,155,157]. To reduce the risk of SUDI, suffocation, strangulation and entrapment, it is advised that pillows, cot bumpers, sheepskins, soft toys and doonas should not be used in young infants’ sleep environments [154,157]. Further, the use of sleep positioners or sleep aids is also not recommended due to these practices or products potentially intensifying rather than reducing the risk of SUDI [154,159]. However, despite these recommendations, many papers in this review found that a considerable proportion of infants were reported by their caregivers to sleep with such items or bedding.

Maintaining a sleep space free of loose or soft bedding and items was explored in 35 of the included review papers. The prevalence of soft items or bedding, in particular, pillow use, varied greatly across studies, with caregiver ethnicity having a strong influence on use. A study investigating practices among population groups internationally demonstrated that families in Hungary (0%), Scotland (4%), Canada (8%) and NZ (9%) reported the lowest incidence of pillow use, with the Chinese sample groups (Hong Kong 80%, Beijing 95% and Chongqing 95%) reporting the highest prevalence [108]. Other studies exploring the use of pillows within an infant’s sleep space reported a prevalence of 14% among an Australian sample [16]; 18% among a UK sample [12]; 19% in an Iranian sample [17]; 21% among an Irish sample [65]; 64% among a Mexican sample [96]; 66% among a UAE sample [36]; 77.3% among a Turkish sample [146]. In a recent Croatian study [45], it was found that 86% of infants slept with a pillow or stuffed toy, whereas in a US study [139], 14% were reported to place their infants to sleep sometimes or always with cot bumpers, pillows or blankets. Many of these studies did not report if pillow type or application was explored, and therefore, it remains unknown if the pillows used within these samples were adult or infant pillows and how these were used (e.g., under the baby’s head or whole body).

An Australian study sought to better understand caregiver views of prevention, treatment and costs of plagiocephaly [101]. Authors found that once the concern or discovery of a flat spot occurred, the majority of parents stopped adhering to the safe sleeping guidelines, with many who were concerned about positional plagiocephaly purchasing various pillows marketed as products to reduce a flat head, frequently initiating use around one-month postpartum. Parents reported that plagiocephaly was more concerning than placing their infant on their back to reduce SUDI risk, as plagiocephaly was more of a reality [101].

Some studies exploring the use of bulky bedding, such as quilts, duvets, comforters and thick blankets, to cover infants during sleep demonstrated that over time, their use has decreased. Colson et al. [60] reported among a cohort of US infants that bulky item use decreased between 1993 and 2000 from 41.2% to 21.6% and again between 2001 and 2010 from 15.9% to 7.8%. Shapiro-Mendoza et al. [127] also reported similar findings, with thick blanket use declining from 56.0% to 27.4% and quilt/comforter coverings declining from 39.2% to 7.9% (using three-year moving averages comparing 1993–1995 to 2008–2010). However, in two recent US studies, it was reported by 39% [47] and 36% [87] of caregivers that they used soft bedding in their infant’s sleep environment. Bombard and colleagues described that the most frequently reported types of soft bedding were bumper pads (19%), plush or thick blankets (18%), pillows (7%), positioners (6%) and stuffed toys (3%). Among a cohort of English infants [12], it was reported that 18% slept under a duvet, and in a sample of UAE caregivers [36], more than 80% used bedding duvets for their infants both in summer and winter. Yikilkan et al. [146] reported that 32% of Turkish infants were placed down to sleep with soft bedding. A recent Australian study found that 21.1% of infants were placed to sleep with soft bedding or items in their sleep environments with items including pillow(s) (8%), doonas/duvets (6.6%), toys (3.5%), sheepskins (2.3%), cushions (1.5%) and bumpers (1.3%) [66].

#### 7.3.4. Infant’s Bed Type or Sleeping Surface

Twenty-seven papers examined the different beds, sleeping places or sleep surfaces babies are placed on during infant sleep. Cots, cribs and bassinets were the most commonly reported bed type. Adult beds were the second most common sleep surface, typically while bed-sharing with an adult caregiver. Other common sleeping surfaces and places were reported to include, but were not limited to, portable or carry cots; Moses baskets; prams or strollers; car seats or baby capsules; bouncers, swings or rockers; co-sleepers, infant nests and snuggle beds; clip-on or side cots; in baby carriers or slings and hammocks and beanbags.

An American study [130], which explored all the sleep surfaces caregivers reported using for their infant at least once during the prior 2 weeks, described findings of 55.2% crib; 38.2% bassinet; 8.7% cradle; 33.2% pack and play; 39.1% adult bed/mattress; 8.0% co-sleeper; 10.2% sofa. A recent study in Victoria, Australia, reported the usual sleep surfaces used in the first 8 weeks postpartum [66]. Bassinets were the most frequently reported sleep location (69.7%), followed by cots (33%), adult beds (17%) and sofas/couches (2.8%); for 6.7% of infants, other sleep surfaces were reported, which included prams, bouncers/rockers, co-sleepers/side beds, being held, car capsules, infant carriers, swings, hammocks, snuggle beds and bean bags [66]. Similar findings were also found among caregivers in Saudi Arabia: 69.8% used a designated baby bed; 9.1% adult bed (alone); 33.8% adult bed (with parents); 6.7% ground; 8.9% car seat; 4.2% couch; 6.7% swing; 5.1% child carrier [11]. Hauck and colleagues [83] measured the proportion of infants aged less than 3 months ever sleeping in an attached co-sleeper (a separate sleep surface adjacent to adult bed) and found the prevalence to be between 3.3 to 5.1%. Shapiro-Mendoza et al. [127] explored the prevalence of soft surfaces or bedding placed under an infant, finding that 29.2% were placed on blankets, 3.7% on cushions, 0.9% pillows, 0.9% waterbed, 0.8% rug, 0.8% sheepskin and 0.3% on a beanbag surface. Shields et al. [21] reported that 10.3% of infants at 1 month were described by their caregivers to sleep only in the parent’s bed and found no real change in this sleeping arrangement by 6 months postpartum (10.2%). The practice of an infant sleeping on an adult bed or mattress was often associated with sharing the sleep surface. The prevalence of bed-sharing and related findings will be further explored in the subsequent section.

#### 7.3.5. Infant Sleep Location: Room-Sharing and Surface Sharing

Eighty-nine papers included in this review explored infant sleep location, measuring the prevalence of infants who room-share and/or bed-share. The proportion of infants who sleep in the same room as their parents or an adult caregiver but did not share a sleep surface (i.e., sleep in their own infant bed) was not always clearly measured or reported. Many papers reporting room-sharing prevalence also included infants who usually bed-shared [20,21,23,45,48,65,72,81,83,135]. Möllborg et al. [104] reported that within a Swedish cohort, 65.7% of infants slept in their own separate bed in the parent’s room. Hutchison et al. [19] reported similar findings from a New Zealand cohort, with 61.3% of infants usually sleeping in the parent’s room in their own bed. Among Thai infants, the incidence of room-sharing but not bed-sharing was 39.3% [39]. A recent American study found that 70.0% of infants room-shared without bed-sharing [119].

Of the 89 papers, 38 described the prevalence of infants who were reported to sleep in a room alone. The prevalence of infants who did not room-share (that is, slept in a room separate to their parents or adult caregiver) varied greatly across included studies, between 0–57%. Hispanic (0%) and Thai (0.14%) infants were reported as the least likely to sleep in a room alone [39,102]. Ford et al., Hutchison et al., Wennergren et al., Barbir et al. and Schluter and Young reported the highest incidence of infants sleeping in a room alone, with a prevalence of 43%, 46%, 46%, 50% and 57%, respectively [15,45,75,88,142]. Among New Zealand infants, Hutchison et al. [19] reported that the prevalence of infants usually sleeping alone in their own room has decreased from 44.6% in 2005 to 30.4% in 2013. An American longitudinal study found that room-sharing fell from 91.9% at hospital discharge to 68.2% by 4 months postpartum [80].

For infants who were reported to bed-share, those that regularly or usually shared a sleep surface ranged from 7.2% within a UK population sample to 89% from an Indian population sample, with 100% of these reporting that their baby had slept beside them on the same bed at one point since birth [44,111]. A recent Malaysian study found that only 47.5% of parents, either always or often, placed their infants to sleep in their own infant bed [122]. For the majority (68.8%) of Malay families, bed-sharing was practiced, with the common reasons for doing so being for ease of breastfeeding and attending to infant needs [122]. A longitudinal study, which examined maternal intention and actual practice at 1 month postpartum, found significantly more mothers reported that they had bed-shared with their infant during the first postpartum month (47.7%) than had said that they planned to (10.7%) [136]. Soothing night-time infant fussiness was reported as the primary reason for the unplanned bed-sharing [136]. Another American longitudinal study reported similar findings, where only 8% intended to bed-share when asked prenatally; however, when asked postnatally, 38% had bed-shared the prior night [84]. In Victoria, Australia, it was found that the bed-sharing prevalence within their sample was 44.7% [66]. Of these, 77.5% reported that they had not planned to bed-share, with explanations for doing so centred on a need to get some sleep and falling asleep by accident [66].

A large Swedish longitudinal population-based study found that the tendency to bed-share correlated with an increased tendency to breastfeed; further, there was a negative association between dummy use and bed-sharing [142]. Similar findings were reported in an Australian study, where it was found infants were approximately twice as likely to still be breastfeed if bed-sharing occurred most or some nights/week and four times more likely if it was ‘normal’ routine [55]. Osberg and colleagues also reported similar findings from a Norwegian study, where 62.7% routinely bed-shared, and of these, 77.2% indicated that bed-sharing was linked to breastfeeding [113].

A recent Australian study found that 33% of participants reported bed-sharing at 0–1 month; however, by 6–12 months, the prevalence of surface sharing had increased to 58% [43]. Conversely, Krouse and colleagues [97] reported that in their longitudinal descriptive study of an American cohort, bed-sharing with an adult caregiver decreased from 47% at one-month-old to 17% at three-months-old, while no mothers had intended to bed-share. The authors further reported that a lack of infant bed was not a factor in bed-sharing among this sample group with all infants having their own bed at 3 months postpartum [97].

Studies differentiating between types of surface-sharing have indicated increasing concern for infants who sofa-share with the risk of SUDI being 67-times that of infants who sleep in their own safe sleeping place [160]. Whether sofa-sharing is intended or not, co-sleeping with an infant on a couch or sofa can be extremely dangerous. Several papers explored the prevalence of infant sleep on a sofa or couch, with the incidence of sofa-sharing found to be as high as 20% among some groups. Crane and Ball [64] reported that 12% of White British mothers stated they had sofa-shared at least once, with 20% stating that they had sofa-shared more than once. Another UK study [12] reported similar findings, with 9.1% of infants ‘ever sofa-sharing’ and four times as many White British mothers (15.8%) reporting falling asleep with their infant on a sofa than Pakistani mothers (4%). Other papers, which also explored sofa-sharing, described findings of 24% of infants ever sharing a sofa or armchair with a caregiver for night-time sleep [16], 10.2% ever sleeping on a sofa in the prior 2 weeks [130], between 12.3% and 17.8% of 2-weeks- to 3-months-old infants ever sleeping on a couch [83] and 19% occasionally sleeping on a couch [79]. Anderson et al. [38] found that occasionally ‘sleeping on a settee’ was reported among 71% of caregivers from a socioeconomically deprived inner-city estate compared to 52% of caregivers from an affluent suburban estate population.

Several studies highlighted caregivers reporting that while they do not usually bed-share, they have admitted to falling asleep on a sofa or recliner while feeding in an effort to adhere to the advice not to bed-share [44,64,130,136]. Tully and colleagues [136] described one mother who reported that she did not bed-share; however, she reported that she routinely co-slept with her infant on a recliner when feeding her baby at night; she recounted that she would fall asleep in the armchair and then wake again there for the next feed. This mother stated ‘I heard about squashing him, [but] if I thought he were in any danger of that, I would lay him back down [in his bassinet]’ [136]. An Australian study reported that 5.8% of mothers reported sofa/couch-sharing with their infants, with the majority (89.7%) of these mothers stating it was not intended to share [66].

#### 7.3.6. Breastfeeding

The World Health Organisation recommends infants are exclusively breastfed for the first 6 months of life to achieve optimal health, growth and development [161]. A recent individual pooled-data meta-analysis concluded that breastfeeding must continue for a minimum of two months to be protective against sudden infant death, with an almost halving of the risk [162]. The longer breastfeeding was preserved, either exclusive or partial, the greater the protective effect for sudden infant death [162]. In total, 59 papers in this review explored breastfeeding practices within the context of safe infant sleep advice.

Between 2–4 months of age, the proportion of infants still receiving any breastmilk was as low as 34.3% in a sample of UK infants to as high as 89% in a New Zealand cohort, with the majority (71%) of these New Zealand infants being exclusively breastfed [44,77]. A recent Spanish study found that a third (34%) of infants aged less than 6 months were no longer breastfed at all [123]. Both Ball et al. [44] and Moon et al. [105] assessed the proportion of infants who never breastfed or did so for less than one week postpartum, with findings reported as 43.6% in the UK cohort and 33.7% in the US cohort, respectively. Norton and Grellner [110] who also measured the proportion of infants in an American cohort who never received any breastmilk or did so for less than one-week postpartum found the prevalence to be as high as 57%.

An Iranian study noted that parent’s education status had a significant association with breastfeeding rates [17]. Similarly, in Australia, Bailey et al. [43] and Cole et al. [55] reported predictors for breastfeeding included maternal education, as well as maternal wellness during pregnancy, birth type, maternal age, marital status, smoking status, in-hospital formula supplementation, admission to a neonatal nursery, dummy use, lack of confidence, lack of knowledge and conflicting advice. Further, they found that infant sleep location had a significant association with breastfeeding. Five studies reported that infants who co-slept were more likely to be exclusively breastfeed at 6 months [43,66,103,104,142]. As described in the previous section, many studies found a significant relationship between bed-sharing and breastfeeding [43,55,66,103,104,133,142].

#### 7.3.7. Summary of Prevalence Findings

Findings demonstrate marked disparities across *all* the key modifiable infant care safe sleeping guidelines and the practices employed by caregivers. Furthermore, findings reveal how diverse the infant care practices being employed by infant caregivers may be across different population groups. These prevalence findings, however, do not provide any understanding as to whether caregivers have received or are aware of the public health safe sleep guidelines and if such advice is therefore being translated into practice.

### 7.4. Awareness of Infant Safe Sleep Messages

Emerging literature suggest that explanations for suboptimal implementation of SUDI risk reduction advice may include ineffective dissemination of current national public health safe sleep messages to parents with a new baby; the large volumes of information received antenatally; culturally inappropriate or insensitive messaging for priority population groups and practical difficulties implementing advice into family routines [116,139,163]. Of papers which met eligibility criteria, 50 explored caregiver awareness of messages promulgated in infant safe sleep campaigns. Many studies also investigated the source from where their information was received.

The extent of caregiver awareness or knowledge varied largely across studies and sample cohorts. The majority, however, reported relatively poor awareness of *all* of the safe sleep advice. For example, in a US study [54] only 40% of caregivers were fully aware of safe sleep facts; in a Malaysian study [122] 44% were categorized as having ‘good knowledge’ (>5 correct out of 9) of SIDS and safe sleeping; a UK study [117] reported that 6.3% did not name any correct SIDS risk reduction strategies, and only 54.3% named three or more; an Australian study [24] found 26% could not correctly identify the six current national safe sleeping messages and a Portuguese cohort [28] of parents found that only seven questions (out of 13) were correctly answered by the majority of respondents.

A New Zealand study, which compared safe sleep message awareness between 2005 and 2013, described an increased awareness among the 2013 caregivers with more 2013 mothers compared with 2005 mothers citing avoiding bed-sharing, keeping the face clear and avoiding soft bedding and room-sharing as SUDI prevention factors [19]. A large Japanese study found that parental awareness of smoke exposure, as a risk factor, and breastfeeding, as a protective factor, were both significantly lower (*p* < 0.001) than awareness of a supine sleep positioning to reduce risk of sudden infant death [85]. In a recent Saudi Arabian study, 64% of caregivers reported not having heard of any SIDS prevention messages; however, despite this, 53.2% correctly stated that babies should be placed on their backs, with only 5.5% stating that babies should be laid on their stomachs [27].

The most common safe sleep advice that papers reported and measured was caregiver awareness of infant sleep position. Of these, the percentage of caregivers reporting correct knowledge of supine sleep ranged between 99.8% at hospital discharge and 97.3% at 4 months post-partum in a US study [80] to as low as 17.9% in a Brazilian study [52]. A recent Brazilian study reported that 82.1% of mothers stated that a baby should sleep in the lateral or ventral decubitus position, with 76.4% of participants indicating they acquired this knowledge from their mothers [52].

The sources from which caregivers gathered or received information about safe infant sleep was varied across studies. In the UK, a qualitative study found that all mothers (*n* = 46) were aware of the UK SIDS-reduction guidelines from leaflets presented to them during their antenatal or postnatal interaction with their healthcare providers [64]. Another UK study (*n* = 400) reported that most mothers recalled a conversation about safe sleep with either a health visitor (62.5%) or midwife (67.3%) [117]. In Portugal, parents (*n* = 204) declared that their main sources of information about SIDS were the internet (53.7%), nurses (34.5%) and doctors (25.9%) [28]. Among a Malay cohort (*n* = 80), it was reported that information was gathered mostly from social media (67.3%) [122]. An American hospital-based safe infant sleep state-wide initiative in Georgia found that 95% of participants (*n* = 420), post-implementation, received their safe sleep information from the hospital [141]. Similarly, an Australian study (*n* = 3341) found that 94% of caregivers reported that they received safe sleep advice from their nurse or midwife [24].

An American study [29] that aimed to increase the promotion of safe sleep advice during antenatal care and conducted a repeated cross-sectional intervention study (*n* = 111, 56 pre- and 55 post intervention) reported only that 11% of pre-intervention women were able to identify all the safe sleep elements, with 32% that reporting their obstetric provider had discussed safe sleep at their 28- or 36-week gestation appointment. Post-intervention knowledge increased to 31% (all safe sleep elements identified), and 78% of women reported their obstetric provider had a discussion about safe sleep [29]. The authors reported that the other sources caregivers indicated receiving their information from were print media (55%), family or friends (45%), hospital staff (34%), online sources (30%), local programs (23%), television (20%), obstetrics provider/staff at previous visit (20%) and different doctor/staff (11%) [29]. Furthermore, they found that 18% reported hearing advice contrary to the American Academy of Pediatrics (AAP) safe sleep recommendations (for example, a grandmother advising prone positioning, a nurse suggesting blankets to prop a baby on its side). Similarly, Eisenberg and colleagues [71] reported findings where the prevalence of any advice received from the family or media was 20–56% for nearly all care practices, with caregivers reporting that advice given was often inconsistent with recommendations. Findings from the analysis of posts in a mother’s Facebook group also demonstrated inconsistency in healthcare provider communication about safe infant sleep [32]. A study of 43 adolescent mothers found that when faced with conflicting advice, many stated that they were most likely to listen to their own mothers over medical professionals [50].

Some papers explored the association between knowledge or awareness of safe sleep advice and the practices implemented. Both Fowler et al. [76] and Cole et al. [24] reported a positive association between knowledge and implementation of safe sleep risk reduction measures. Similarly, Walcott et al. undertook logistic regression analyses, which found that the receipt of information was significantly correlated with the implementation of safe sleep behaviours [141]. However, a study by Caraballo and colleagues [50] demonstrated the opposite, in that despite caregiver knowledge of safe sleep recommendations, adherence with advice was poor. Chung-Park [54] reported similar findings, where 85% indicated that supine positioning was safest; however, of those, only 69% also practiced supine positioning. Further, they reported that it was found that parents’ opinions of sleep position and their practices were significantly associated (*p* < 0.001), whereas knowledge on safe sleep facts and their practices was not (*p* = 0.611). Likewise, findings from a study by Raines [33] indicated that although new parents may know about safe sleep recommendations, their actions and behaviours in the home are not consistently guided by that knowledge but, rather, by other influential factors, such as opinions and advice shared by grandmothers or fathers/husbands.

Findings in the 50 papers which were identified to have explored caregiver receipt and awareness of safe infant sleep advice demonstrate that knowledge of SUDI risk reduction is diverse internationally, and similar to findings in the prevalence data, they are largely dependent on the population group being sampled. These findings, however, do not explore or explain why there may be a disconnect between caregiver knowledge of safe sleep advice and the practices they adopt.

### 7.5. Challenges Implementing Safe Sleep Advice

Understanding common concerns, barriers and/or challenges families encounter when implementing safe sleep advice, together with facilitators that support consistent uptake of current guidelines, is essential for effective safe sleep messaging and to ultimately provide supportive and practical advice. This subject was explored by 52 articles included in this review. A mixed methods or qualitative design was utilised by the majority of studies that collected data exploring barriers and/or challenges, as well as the facilitators caregivers encounter when employing safe sleep advice. Bed-sharing practices; infant sleep positioning and soft sleep surfaces, bedding or pillow use were the most common infant care practices explored by these papers.

Some studies highlighted similarities and differences in perceptions for safe sleep practices amongst different cultural groups and how practices also differed across ethnicities. For instance, a study by Moon and colleagues [105] found the African-American mothers in their cohort perceived the link between safe sleep practices and SIDS implausible. Conversely, a Canadian study [30] found that mothers in their sample acknowledged infant sleep arrangements being related to SIDS risk. Mathews et al. [102] found that although African-American and Hispanic parents had similar concerns, their adopted infant care practices differed. They suggest that this finding may help explain some of the racial/ethnic disparities seen in sleep-related infant deaths. Similarly, a study exploring the adoption of SIDS-reduction guidance in a UK bi-cultural community also found that cultural circumstances and values largely influenced parenting decisions and practices [64]. They reported that Pakistani mothers tended to prioritise their traditional practices and dismiss official guidance, and White British mothers consciously adapted or dismissed several aspects of the advice to suit their preferred parenting decisions and personal circumstances.

Recent studies have demonstrated that family social networks and social norms are associated with engagement in many infant care practices. Colson et al. [61] reported that subjective social norms and attitudes were most strongly associated with behaviours heavily influencing practice. Moon and colleagues [31] also found strong associations between social network norms and all the measures of sleep practices they examined. Further, they found that caregivers with more exclusive social networks (kin-centric) had a stronger influential role in terms of their effect on caregiver practices compared to caregivers with expansive (friend-based) social networks. A study by Cornwell et al. [62], who also found that social networks play a significant role in infant sleep practices adopted during the first few months of an infants’ life, reported that mothers will more likely change sleep practices when health professionals and/or family members advise them to and that for mothers who initially follow safe practices, their probability of change increased if their network members substantially espoused unsafe practices. One study, which utilised photovoice methodology, found that the main facilitators contributing to safe sleep practices were furnishing, knowledge, routines, social support and visual reminders [35]. Similarly, Lau and Hall [30] described four factors that mothers reported experiencing regarding infant sleep safety: perceptions of everyone’s needs, familial influences, attitudes and judgements from outsiders and resource availability and accessibility.

The common facilitators, concerns or challenges reported in papers regarding the uptake of sleep position advice included safety reasons, fear of choking, plagiocephaly, infant preference, sleep quality and comfort [34,37,51,54,101]. The reasons reported for bed-sharing were consistent across the papers that explored this practice. Reasons centred around the mother–infant dyad getting the best possible sleep; enabling easier, more convenient feeding throughout the night; perceived safety and protection and being viewed as a cultural caring behaviour promoting comfort and closeness [41,42,78,86,147]. One study reported that a small number of parents practised bed-sharing as they did not have a cot [76]. Tully and colleagues [136] explored reasons for unplanned bed-sharing and found that the primary reason was to soothe night-time infant fussiness. Factors influencing decisions by parents regarding the use of soft bedding and sleep surfaces for their infants have been examined [25,26]. They reported that the primary reason for using soft surfaces and soft bedding (including bumper pads) was infant comfort, perceived safety and aesthetics. Pillow use was most often to make the sleep surface softer or more comfortable, followed by infant safety: when infants slept on beds or sofas, pillows were used to create a barricade to prevent falling [26]. Martiniuk et al. [101] sought to understand caregiver views of plagiocephaly and found that parents were ‘willing to do anything’ to prevent plagiocephaly, including using products or sleeping positions that are contraindicated with SUDI risk reduction guidelines.

The 52 papers, which have provided some insight into caregiver engagement with safe sleep advice and the implementation of this advice, demonstrate that each individual family’s experiences, circumstances and perspectives largely influences their infant care practices. Understanding the barriers and facilitators at a population-level presents a great challenge, as revealed by these studies.

## 8. Discussion of Review Findings

Understanding how families receive and adopt safe sleep public health advice, together with the contemporary infant sleep practices adopted by families with young infants, holds a strong global interest, as evidenced by the large number of studies conducted internationally. This review explored findings from 137 articles which investigated infant caregiver engagement with safe infant sleep recommendations. Research exploring how infants are cared for among the broader population in which most infants grow, develop and thrive provides vital information to better understand and contextualise circumstances in which infants die suddenly and unexpectedly.

The demographics of participants in the included articles varied across study populations by ethnicity, marital status, maternal age, infant age, education level and socioeconomic status. Studies that examined and analysed practices and behaviours employed by infant caregivers from various groups found a variety of disparities across population groups. Cultural values and ethnicity have a strong influence on practices employed and the adoption of safe sleep messages. For example, shared sleep with a caregiver is a culturally valued practice within many Australian First Nations families [164]. Risk minimisation approaches that provide practical strategies for creating safer shared sleeping environments relevant to family values and circumstances are more likely to engage caregivers than risk elimination messages that advise that caregivers should never bed-share [64,66]. Caution should, however, be taken in generalising findings from one specific region or cohort of caregivers; rather, an intentional or targeted exploration of priority population groups or specific geographical locations experiencing higher infant mortality should be undertaken to better inform future campaigns and strategies.

Recent studies examining SUDI rates in Queensland, Australia, reported Aboriginal and Torres Strait Islander SUDI mortality rates to be over 3.5-times those of non-Indigenous Australian infants [164,165]. Shipstone and colleagues [164] advise that to redress the inequities between the two groups, a detailed understanding of Aboriginal and Torres Strait Islander populations is required to design and deliver appropriate initiatives, in collaboration with Aboriginal eldership and governance. Similarly, Crane and Ball [64] described within their UK study that safe sleep information does not meet the needs of immigrant families and concluded that tailored information, acknowledging cultural differences in infant care practices, is vital. Such conclusions are likely relevant in other nations facing disparities in infant mortality outcomes for priority populations. It remains unknown how and if public health safe sleep campaign messages are being received among population groups at greater risk.

Infant age varied across included studies, with infants reported to be as young as days old through to 3 years of age, with some articles not reporting infant age at all. Comparing prevalence across this range is limited, due to changes and developmental milestones children experience as they grow, which significantly influence the practices caregivers adopt when providing care for their infant. Many of the included longitudinal studies identified shifts in caregivers’ practices as their infants grew [21,90,93]. Consequently, when investigating infant care and sleep practices, care must be taken to closely consider the infants age and development due to the strong influence this may play in the practices employed and challenges families encounter.

While randomised controlled trials (RCT) are regarded the gold standard for causal inference, common limitations reported for the use of the RCT methodology for SUDI studies include ethical reasons, generalisability of findings and the infrequency of events within the population, thus requiring an impractically large sample size [166,167,168]. Consequently, the majority of contemporary SUDI studies are observational [166], consistent with this review, where most studies applied cross-sectional survey designs. Most studies used convenience or purposive sampling strategies; few studies defined robust sampling frames [16,85,131,142] in which the target population denominators were clearly defined.

Study data collection methods mostly relied on self-report data. Stigma around caregiver use of unsafe sleep practices may arguably influence caregiver reports, resulting in the underreporting of unsafe sleep practices [49,64,124,151]. Crane and Ball [64] described mothers reporting the use of unsafe sleeping environments despite being advised by health professionals against them. Cole and colleagues [25] reported similar findings. When analysing prevalence data, this behaviour needs to be carefully considered, as the incidence of certain ‘unsafe’ practices could very well be greater than what is reported.

### Limitations

This review holds several limitations. Firstly, the focus of the scoping review was to provide breadth rather than depth of information on a particular topic. As scoping review methods do not usually follow a formal process of methodological appraisal, in which study quality is critiqued [5,7], the interpretation of evidence is limited. Grey literature is also excluded. Given the volume of eligible studies identified by the search strategy, for pragmatic considerations, this review was limited to peer-reviewed empirical studies, which were also determined to address the aims of the review. Further, eligibility criteria limited publications to the English language; thus, potentially relevant studies in other languages were excluded. Eligibility criteria also did not define or limit the age of infants from when prevalence data were measured or collected.

## 9. Conclusions

This review has comprehensively scoped the peer-reviewed published literature on primary infant caregiver engagement with infant safe sleep campaign messages and practices related to caregiver infant sleep practice. Empirical literature, benchmarking the prevalence of infant care practices, together with studies that have investigated caregiver knowledge and awareness of current public health program messages and emerging concerns related to the implementation of infant safe sleep practices have been summarised. Findings from this review demonstrate a need for careful and ongoing exploration of infant sleep practices and the family’s receipt of safe sleep advice. While prevalence data cannot ascertain interventions or strategies to reduce sudden infant death, they can identify disparities and reveal population groups that may need targeted education or support.

## Figures and Tables

**Figure 1 ijerph-19-07712-f001:**
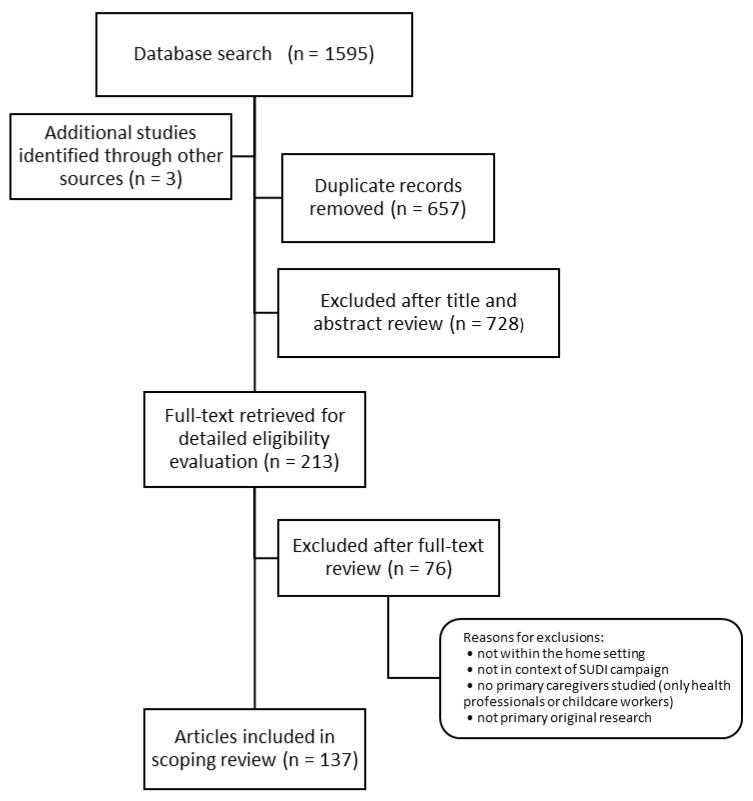
Study selection (PRISMA Flowchart).

**Table 1 ijerph-19-07712-t001:** Search strategy string.

Population	Sample cohort terms	Caregiver* OR parent* OR mother* OR maternal OR women
		AND
	Study type terms	“cross-sectional” OR survey OR questionnaire OR interview
		AND
Context	SUDI terms	sids OR sudi OR suid OR “sudden infant death” OR “unexplained infant death*” OR “sleep-related death*” OR “cot death*” OR “crib death*”
		AND
Concept	Uptake and engagement with safe sleep advice (i.e., prevalence of care practices, awareness of messages, challenges implementing advice)	prevalence OR “infant care” OR practice* OR knowledge OR awareness OR understand* OR recommendation* OR advice OR message* OR challeng* OR barrier* OR difficult* OR facilitator* OR enabler* OR concern*

Limits: Dates January 2000 to May 2021; Language English. N.B. * represents truncation.

**Table 2 ijerph-19-07712-t002:** Eligibility criteria.

**Inclusion Criteria**
Peer-reviewed original primary research Participants and sample groups were *current primary infant caregivers* (e.g., mothers, fathers, grandparents) Assessment of study outcomes occurred when the caregiver was caring for their infant *within a home environment* Study measures or outcomes were within the *context of national public health campaigns targeted at reducing sleep-related infant mortality* through safe sleep messages and/or SUDI risk reduction advice Studies which identified key outcomes of interest through intentions to: (a) monitor and/or measure known key modifiable sleeping or care practices promulgated in current SUDI risk-reduction messages; (b) explore primary caregiver awareness or knowledge of the key safe sleeping messages or advice and/or (c) explore facilitators and challenges or barriers caregivers face when engaging with safe sleep advice in their home environments. (NB: For the studies where multiple outcomes were examined, only barriers and facilitators, knowledge and awareness and/or prevalence data relating to key infant care practices associated with safe infant sleep messages were extracted and reported) Prospective, cross-sectional and/or longitudinal study designs, including both qualitative and quantitative data Published between January 2000 and May 2021
**Exclusion Criteria**
Irrelevant exposure, which was not measuring infant care practice in relation to sleep-related infant mortality, specifically SUDI risk-reduction campaigns or safe infant sleep messages No key outcomes of interest reported, i.e., the study examined caregiver intention of practice or assessed factors that may be associated with sleep-related infant mortality but did not measure any prevalence of usual or actual practice) (NB: where studies assessed multiple outcomes (for example, attitudes towards care practices, awareness of safe sleep advice and prevalence of infant care practices, then, where relevant, awareness and prevalence data were extracted from the paper for the purposes of this review; if studies only related to caregiver attitudes toward care practices or decision-making without any awareness measures, prevalence findings or possible barriers/facilitators, these studies were excluded) Full article not available in English

**Table 3 ijerph-19-07712-t003:** Target outcome measures charted.

	**Prevalence**								**Awareness**	**Challenges**
**Article**	**Sleep Position**	**Smoking**	**Items in Sleep Area**	**Bed Type or Surface**	**Room Location**	**Surface Sharing**	**Breast Feeding**	**Other Practice**		
Abdulrazzaq et al. (2008) [36]	●	●	●		●	●				
Aitken et al. (2016) [37]	●			●						●
Ajao et al. (2011) [26]										●
Alahmadi et al. (2020) [11]	●	●	●	●	●	●	●	●	●	
Alzahrani et al. (2020) [27]									●	
Anderson et al. (2002) [38]	●	●			●	●	●	●		●
Anuntaseree et al. (2008) [39]	●				●	●				
Ateah and Hamelin (2008) [40]						●	●		●	●
Austin et al. (2017) [41]						●				
Baeis et al. (2015) [17]	●	●	●		●	●	●			
Bailey (2016) [42]						●	●			●
Bailey et al. (2020) [43]						●	●			●
Ball et al. (2012a) [12]	●	●	●	●	●	●	●	●		
Ball et al. (2012b) [44]		●				●	●	●		
Barbir et al. (2020) [45]	●	●	●			●			●	
Beck et al. (2002) [46]	●	●					●			
Bombard et al. (2018) [47]	●		●	●		●				
Brenner et al. (2003) [48]	●	●		●	●	●		●		
Broussard et al. (2012) [49]	●	●				●	●			
Caraballo et al. (2016) [50]		●				●			●	●
Cesar et al. (2018) [51]	●								●	
Cesar et al. (2019) [52]	●								●	●
Chung et al. (2003) [53]	●	●					●			
Chung-Park (2012) [54]	●			●					●	●
Cole et al. (2020) [16]	●	●	●	●	●	●	●	●		
Cole et al. (2020) [55]	●	●			●	●	●	●		
Cole et al. (2021) [24]	●	●	●		●	●	●		●	
Cole et al. (2021) [25]	●	●	●		●	●	●			●
Colson et al. (2000) [56]	●								●	●
Colson et al. (2001) [57]	●								●	●
Colson et al. (2006) [58]	●									●
Colson et al. (2009) [59]	●			●						●
Colson et al. (2013) [60]			●		●	●				●
Colson et al. (2017) [61]	●							●	●	
Cornwell et al. (2021) [62]	●					●				
Corwin et al. (2003) [63]	●									
Crane and Ball (2016) [64]	●		●	●	●		●		●	●
Cullen et al. (2000) [65]	●	●	●		●			●	●	
Cunningham et al. (2018) [66]	●		●			●	●			
da Silva et al. (2019) [67]	●					●				●
Damato et al. (2016) [13]	●	●	●	●	●	●	●			
Douglas et al. (2001) [68]	●	●			●	●	●		●	
Duzinski et al. (2013) [69]	●		●	●		●				
Efe and Ak (2012) [70]	●	●				●		●		
Eisenberg et al. (2015) [71]	●					●	●		●	
Epstein and Jolly (2009) [72]	●	●			●	●			●	●
Erdoğan and Turan (2018) [73]	●	●		●		●				
Erick-Peleti et al. (2007) [74]		●								
Fernandes et al. (2020) [28]									●	
Ford et al. (2000) [75]	●		●		●	●				
Fowler et al. (2013) [76]	●	●	●	●		●			●	●
Galland et al. (2014) [77]	●	●				●	●	●		
Gaydos et al. (2015) [78]	●					●			●	●
Gibson et al. (2000) [79]	●	●		●		●	●			●
Goodstein et al. (2015) [80]	●			●	●			●	●	
Haas et al. (2017) [18]	●	●	●		●	●	●	●		
Hamadneh et al. (2016) [81]	●	●	●			●		●		
Hannan et al. (2020) [82]	●	●					●			
Hauck et al. (2008) [83]	●			●	●	●				●
Hauck et al. (2015) [84]	●	●		●		●		●	●	
Hirabayashi et al. (2016) [85]	●	●					●		●	
Homer et al. (2012) [86]						●				●
Hussain et al. (2018) [87]	●	●	●			●	●			
Hutchison et al. (2006) [88]	●	●			●	●	●	●	●	●
Hutchison et al. (2007) [89]	●							●		●
Hutchison et al. (2010) [90]	●	●			●	●	●	●	●	●
Hutchison et al. (2015) [19]	●	●	●		●	●	●	●	●	
Hwang et al. (2013) [91]	●	●					●			
Hwang et al. (2016) [92]	●			●	●		●			
Inbar et al. (2005) [93]	●									
Joyner et al. (2016) [94]								●	●	●
Kelmanson (2013) [95]	●			●						
Kihlström et al. (2020) [35]								●		●
Konstat-Korzenny et al. (2019) [96]	●		●			●		●		
Krouse et al. (2012) [97]						●			●	●
Kuhlmann et al. (2016) [29]									●	●
Lahr et al. (2005a) [98]		●				●				
Lahr et al. (2005b) [99]	●									
Lahr et al. (2007) [100]	●	●				●	●			
Lau and Hall (2016) [30]									●	●
Martiniuk et al. (2016) [101]	●		●						●	●
Mathews et al. (2015) [102]	●	●				●	●		●	●
Miladinia et al. (2015) [103]	●	●				●	●			
Möllborg et al. (2011) [104]	●					●	●	●		
Moon et al. (2010) [105]	●	●			●	●	●	●		●
Moon et al. (2019) [31]										●
Moon and Omron (2002) [106]	●					●		●		●
Nelson et al. (2001a) [107]	●	●							●	
Nelson et al. (2001b) [108]			●		●	●			●	
Nelson et al. (2005) [109]							●	●	●	
Norton and Grellner (2011) [110]						●	●			
Nongkynrih et al. (2017) [111]	●		●	●		●			●	
Oden et al. (2012) [112]	●							●		
Osberg et al. (2021) [113]	●	●				●	●			
Panaretto et al. (2002) [114]	●	●			●	●	●			●
Paterson et al. (2002) [115]		●				●	●			●
Pease et al. (2017) [116]		●							●	●
Pease et al. (2018) [117]							●		●	
Phares et al. (2004) [118]	●	●					●	●		
Pretorius et al. (2020) [32]									●	●
Provini et al. (2017) [119]	●	●			●	●	●			
Raines (2018) [33]										●
Roberts and Upton (2000) [120]	●	●							●	
Robida and Moon (2012) [121]	●								●	
Rohana et al. (2018) [122]	●					●			●	
Ruiz Botia et al. (2020) [123]	●	●				●	●	●		
Salm Ward and Ngui (2015) [124]	●					●	●			●
Salm Ward et al. (2018) [125]	●					●	●			
Salm Ward et al. (2016) [126]						●				●
Sawaguchi et al. (2002) [20]	●	●	●		●	●	●	●		
Schluter et al. (2007) [14]	●	●	●	●	●	●	●	●		
Schluter and Young (2002) [15]	●	●	●	●	●	●	●	●		
Shapiro-Mendoza et al. (2015) [127]	●		●	●		●				
Shields et al. (2005) [21]	●	●		●	●	●	●			
Sivan et al. (2004) [128]	●									
Smith et al. (2012) [129]	●	●								●
Smith et al. (2016) [130]				●		●	●		●	
Smylie et al. (2014) [131]	●	●					●			●
Specker et al. (2020) [23]	●	●	●		●	●	●			
Sperhake et al. (2009) [132]	●								●	
Strömberg Celind et al. (2017) [133]	●	●				●	●	●		
Tipene-Leach et al. (2010) [134]	●	●			●	●	●	●	●	●
Tirosh et al. (2000) [135]	●				●	●				
Tully et al. (2015) [136]			●	●	●	●			●	●
van Sleuwen et al. (2003) [137]	●	●	●		●	●		●		
Vernacchio et al. (2003) [138]	●									
Varghese et al. (2015) [34]									●	●
Vilvens et al. (2020) [139]	●		●			●				
Von Kohorn et al. (2010) [140]	●									●
Walcott et al. (2018) [141]	●					●	●		●	
Wennergren et al. (2021) [142]	●		●			●		●		
Willinger et al. (2000) [143]	●									●
Wilson (2000) [22]	●	●		●	●	●	●	●		
Woods et al. (2015) [144]	●					●				●
Wright et al. (2014) [145]	●									●
Yikilkan et al. (2011) [146]	●	●	●			●	●	●	●	
Zoucha et al. (2016) [147]						●			●	●

NB. Outcome measures reported in papers are indicated with a ●.

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
