# Peer review of "Infant Care Practices, Caregiver Awareness of Safe Sleep Advice and Barriers to Implementation: A Scoping Review"

_ijerph, 2022, doi:10.3390/ijerph19137712_

Round 1

Reviewer 1 Report

This is a relevant and well written script. The content is comprehensive and presents a current review of the literature.

Minimal corrections required:

L51 'mortality' missing?

L122 'at' not 'as'

Table 2: presume the formatting will be easier to read int he published version. All content in the table is clear.

Prisma diagram, check figures presented.

Author Response

Response to Reviewer 1 Comments

This is a relevant and well written script. The content is comprehensive and presents a current review of the literature.

Thank you for your comment.

Minimal corrections required:

L51 'mortality' missing?

Response: Thank you. This has been added to the manuscript

L122 'at' not 'as'

Response: Thank you. We have corrected this in the manuscript

Table 2: presume the formatting will be easier to read in the published version. All content in the table is clear.

Response: Thank you we have revised the formatting.

Prisma diagram, check figures presented.

Response: Thank you for identifying this error. The ‘duplicate records removed’ amount has been corrected to (n=657) from the incorrect number of (n=654).

Reviewer 2 Report

Thank you for your valuable research that could contribute to the advancement of this field. I have read it with great interest. I would like to ask you a few questions for a better understanding.

Introduction

Point 1
.
The significance of this study would be more convincing if specific figures were shown that
the safe-related infant mortality incidence has plateaued over the past two decades = but many deaths due to SUDI and other causes are still occurring.

Methods

Point
2.
What is the definition of "national public health campaigns" in the eligibility criteria in
Table2.? Were there any papers from countries excluded because they did not have a SUDI national campaign? Or, if eligibility criteria are met if any of the six target outcomes (Australian Safe Sleep campaign messages) are included, would it be more straightforward and clearer to specify
those six outcomes(practices)?

Point3.

Why did you not remove studies with unknown timing of survey implementation?
Since safety-related infant mortality rates have plateaued over the past 20 years, I thought you were covering papers published since 2000?

Results:

P
oint 4.
Wha
t does "Additional studies identified through other sources (n=3)" in Figure 1. mean? If there are three additional studies, the number of "Full-text retrieved for detailed eligibility evaluation" should be n-216.

P
oint 5.
The results for
each country are very interesting. However, it would be helpful to obtain useful information for readers if they were more comparable. For example, how about adding country names or areas to Table 3? It would be helpful to see the difference in the prevalence of practice by country and consider the cultural background influence.

P
oint 6.
Both awareness and knowledge
seem to be included in the "7.4. Awareness of Infant Safe Sleep Messages" section. Was it difficult to separate the two, or was the purpose to review both?

Point 7.

It would be good to know which papers are qualitative or quantitative
, although it may be difficult to list them in Table 3 due to space limitations.

Discussion.

Point
8.
For example, one of the main findings
of this study is that there were diverse disparities among groups and nations.
I would be interested to hear
if there was anything universal in the findings or what kind of research the authors think is needed in the future.

Point
9.
You mention that the Study data collection methods mostly relied on self
-reported data. What were some data collection methods that did not rely on self-reporting?

Point 1
0.
As noted in the limitations, why didn't you take into account the age of
frequent SIDS and focus only on infants, for example? Breastfeeding and co-sleeping with the child will naturally decrease after the age of 1 year. Focusing on the age at which SIDS should be prevented the most would have been clearer regarding parental awareness and barriers.

P
oint 11.
In the end, what are the most important finding
s of this study? What research is needed to be done in this area in the future

Author Response

Response to Reviewer 2 Comments

Thank you for your valuable research that could contribute to the advancement of this field. I have read it with great interest. I would like to ask you a few questions for a better understanding.

Introduction
Point 1.
The significance of this study would be more convincing if specific figures were shown that the safe-related infant mortality incidence has plateaued over the past two decades = but many deaths due to SUDI and other causes are still occurring.

Response: Thank you for this observation. We have provided a reference by Muller-Nordhorn and colleagues (2020) BMC Pediatrics which demonstrates international time trends in sudden infant death, 1969-2012 and a relevant Australian data from the Australian Bureau of Statistics to support this statement.

Methods
Point 2.
What is the definition of "national public health campaigns" in the eligibility criteria in Table2.? Were there any papers from countries excluded because they did not have a SUDI national campaign? Or, if eligibility criteria are met if any of the six target outcomes (Australian Safe Sleep campaign messages) are included, would it be more straightforward and clearer to specify those six outcomes(practices)?

Response: Thank you for your query. As the purpose of this scoping review was to explore literature relating to family engagement with infant safe sleep public health messages a criteria for inclusion needed to ensure that the study was reporting on findings in relation to safe infant sleep messages to reduce SUDI, not just prevalence of infant care practices in general. Some of the key safe sleep messages and therefore target prevalence outcomes (such as keep baby smoke free or breastfeed baby) are also common health promotion messages for many other healthy infant programs.

Point3.
Why did you not remove studies with unknown timing of survey implementation? Since safety-related infant mortality rates have plateaued over the past 20 years, I thought you were covering papers published since 2000?

Response: Thank you for this comment. To clarify, articles were only included if they were published since the year 2000. This twenty year period was selected to provide the most recent and relevant publications relating to the research question. We did not exclude papers that did not define the data collection period if this key eligibility criteria was met (published since 2000) in order to avoid the risk of exclusion of relevant, eligible studies.

No change made to text.

Results
Point 4.
What does "Additional studies identified through other sources (n=3)" in Figure 1. mean? If there are three additional studies, the number of "Full-text retrieved for detailed eligibility evaluation" should be n-216.

Response: Thank you for identifying this error. The ‘duplicate records removed’ amount should have been (n=657). This has been corrected.

Point 5.
The results for each country are very interesting. However, it would be helpful to obtain useful information for readers if they were more comparable. For example, how about adding country names or areas to Table 3? It would be helpful to see the difference in the prevalence of practice by country and consider the cultural background influence.

Response: Thank you for this feedback and suggestion. While we agree it would be helpful for this information to be also included in Table 3 due to the length and detail already contained in this table we have left the table unchanged. This information is however already contained in the Supplementary Material Table S1.

Point 6.
Both awareness and knowledge seem to be included in the "7.4. Awareness of Infant Safe Sleep Messages" section. Was it difficult to separate the two, or was the purpose to review both?

Response: Thank you for your comment. Separating and analysing message awareness independently of caregiver knowledge in many instances within the articles was not possible. Where caregiver knowledge was measured, this usually indicated a degree of awareness they had with the messages and therefore where knowledge was measured this was deemed as exploring caregiver awareness in greater detail.

Point 7.
It would be good to know which papers are qualitative or quantitative, although it may be difficult to list them in Table 3 due to space limitations.

Response: Thank you for this feedback and suggestion. Again while we agree it would be helpful for this information to be included in Table 3, due to the length and detail already contained in this table we have left the table unchanged. This information is contained in the Supplementary Material Table S1.

Discussion
Point 8.
For example, one of the main findings of this study is that there were diverse disparities among groups and nations.
I would be interested to hear if there was anything universal in the findings or what kind of research the authors think is needed in the future.

Response: Thankyou for this observation. Summary information relating to review findings for specific practices has been addressed in the challenges in implementing safe sleep advice section. Specific recommendations relating to the need for tailored information acknowledging cultural differences in infant care practices was included in the conclusion. Summary information relating to review findings for specific practices was addressed in the challenges in implementing safe sleep advice section.

Point 9.
You mention that the Study data collection methods mostly relied on self-reported data. What were some data collection methods that did not rely on self-reporting?

Response: Thank you for this observation. Section 7.1.1 discusses the results of the data collection methods identified in the eligible articles (n=137). Some studies utilised a mixed methods design; in addition to self reported data, some included a social media content analysis, focused ethnography and sentinel data collection.

Point 10.
As noted in the limitations, why didn't you take into account the age of frequent SIDS and focus only on infants, for example? Breastfeeding and co-sleeping with the child will naturally decrease after the age of 1 year. Focusing on the age at which SIDS should be prevented the most would have been clearer regarding parental awareness and barriers.

Response: This was a scoping review with the aim to explore the breadth and extent of literature that relate to family engagement with safe sleep public health messages relevant to infants (0-12 months); we therefore did not limit infant age to the period when SIDS is most prevalent (0-6 months). Further, understanding how an infant’s developmental milestones impact the practices caregivers adopt to accommodate the rapid developmental changes in the first year of life was of interest in this scoping review.

No changes made.

Point 11.
In the end, what are the most important findings of this study? What research is needed to be done in this area in the future

Response: Thank you for your comment. As highlighted in the abstract and conclusion, this scoping review has demonstrated a need for ongoing monitoring of infant sleep practices and family engagement with safe sleep advice so potential population and disparate groups at greater risk can be identified and targeted strategies and resources developed for these priority populations.

Reviewer 3 Report

Thanks for giving me the opportunity to review your paper. I really appreciate the extensive background work and research that must have gone into writing this scoping article. Here are my comments, 

1. Review of the literature is up to date and very well done. Article is complete and addresses 2 decades of literature which is a feat in itself. 

2. It took me 6 hours to read through the article, and I may not be a fast reader, so I would recommend decreasing the length of the article in possible 2 ways - 

---to concise and remove repeated informations wherever possible. 

---Add up to 2-3 tables, flowcharts or pictures, that can replace a lot of text and will make it more readable for our millennial crowd. 

To summarize I feel the article is very extensive and well done, just need to shorten it a bit. 

Thanks and all the best. 

Regard.

Author Response

Response to Reviewer 3 Comments

Thanks for giving me the opportunity to review your paper. I really appreciate the extensive background work and research that must have gone into writing this scoping article. Here are my comments, 

1. Review of the literature is up to date and very well done. Article is complete and addresses 2 decades of literature which is a feat in itself.

Thank you for your kind comment

2. It took me 6 hours to read through the article, and I may not be a fast reader, so I would recommend decreasing the length of the article in possible 2 ways - 

---to concise and remove repeated information wherever possible. 

---Add up to 2-3 tables, flowcharts or pictures, that can replace a lot of text and will make it more readable for our millennial crowd. 

To summarize I feel the article is very extensive and well done, just need to shorten it a bit. 

Response: This scoping review is understandably comprehensive as it addresses three inter-related issues in the current literature related to infant care practices, namely a) prevalence of practices associated with the six key modifiable practices and b) awareness of safe sleep messages and c) caregiver challenges with safe sleep messages. Findings were synthesised and presented as summaries where possible. Awareness of, and challenges with, safe sleep recommendations were included in this review as these factors directly impact decision making by caregivers relating to uptake of safe sleep advice and infant care practices they use for their baby. Selected data only is presented in the results, with more detailed information presented in Supplementary Material Table S1.

Text has been reduced throughout where possible to make presentation more succinct.

Reviewer 4 Report

The aim of this study was to conduct a scoping review of the existing literature related to safe sleep messaging for infants. This manuscript addresses and important area, how caregivers implement safe sleeping messaging to reduce SUDI risk, and is of interest to the readership of the International Journal of Environmental Research and Public Health.

This is manuscript is well-written and the research gap and aims are clearly stated. However, prior to publication the authors should make a few minor edits to improve the readability of the Table 2 and Table 3. 

Table 2: Authors may consider remove the bullet points and left justifying all content (if permitted by the journal) to improve readability. In addition, ‘Inclusion criteria’ and ‘Exclusion criteria’ could be bolded. 

Table 3: Authors may consider reformatting as the table header is unreadable. It would also be nice to add this same header to the top of each page as this table crosses three journal pages.

Author Response

Response to Reviewer 4 Comments

The aim of this study was to conduct a scoping review of the existing literature related to safe sleep messaging for infants. This manuscript addresses and important area, how caregivers implement safe sleeping messaging to reduce SUDI risk, and is of interest to the readership of the International Journal of Environmental Research and Public Health.

This is manuscript is well-written and the research gap and aims are clearly stated.

Thank you for your comment.

However, prior to publication the authors should make a few minor edits to improve the readability of the Table 2 and Table 3. 

Table 2: Authors may consider remove the bullet points and left justifying all content (if permitted by the journal) to improve readability. In addition, ‘Inclusion criteria’ and ‘Exclusion criteria’ could be bolded. 

Response: Thank you for your suggestions, we have revised the formatting for this table.

Table 3: Authors may consider reformatting as the table header is unreadable. It would also be nice to add this same header to the top of each page as this table crosses three journal pages.

Response: Thank you for your feedback and we have adjusted the header formatting for Table 3.